



# Influence of cloud microphysical processes on black carbon wet removal, global distributions, and radiative forcing

Jiayu Xu[1,‖], Jiachen Zhang[2,‖], Junfeng Liu[1,*], Kan Yi[1], Songlin Xiang[1], Xiurong Hu[1], Yuqing Wang[1], Shu Tao[1] and George Ban-Weiss[2]

[1] Laboratory for Earth Surface Processes, College of Urban and Environmental Sciences, Peking University, Beijing, China

[2] Department of Civil and Environmental Engineering, University of Southern California, Los Angeles, CA, USA

[*] Correspondence to: J. Liu (jfliu@pku.edu.cn)

[‖] J. Xu and J. Zhang contributed equally to this paper.

**Abstract.** Parameterizations that impact wet removal of black carbon remain uncertain in global climate models. In this study, we enhance the default wet deposition scheme for BC in the Community Earth System Model (CESM) to (a) add relevant physical processes that were not resolved in the default model, and (b) facilitate understanding of the relative importance of various cloud processes on BC distributions. We find that the enhanced scheme greatly improves model performance against HIPPO observations relative to the default scheme. We find that convection scavenging, aerosol activation, ice nucleation, evaporation of rain/snow, and below cloud scavenging dominate wet deposition of BC. BC conversion rates for processes related to in-cloud water/ice conversion (i.e., riming, the Bergeron processes, and evaporation of cloud water sedimentation) are relatively smaller, but have large seasonal variations. We also conduct sensitivity simulations that turn off each cloud process one at a time to quantify the influence of cloud processes on BC distributions and radiative forcing. Convective scavenging is found to most significantly influence BC concentrations at mid-altitudes over the tropics and even globally. In addition, BC is sensitive to all cloud processes over the Northern Hemisphere at high latitudes. As for BC vertical distributions, convective scavenging has a dominant influence. Aerosol activation mainly increases the fraction of column BC below 5 km





whereas ice nucleation decreases that above 10 km. During wintertime, the Bergeron process also

significantly increases BC concentrations at lower altitudes over the Arctic. Our simulation yields a

global BC burden of 85 Gg; corresponding direct radiative forcing (DRF) of BC estimated using the

Parallel Offline Radiative Transfer (PORT) is 0.13 W m$^{-2}$, much lower than previous studies. The range

of DRF derived from sensitivity simulations is large, 0.09-0.33W m$^{-2}$, corresponding to BC burdens

varying from 73 Gg to 151 Gg. Due to differences in BC vertical distributions among each sensitivity

simulation, fractional changes in DRF (relative to the baseline simulation) are always higher than

fractional changes in BC burdens; this occurs because relocating BC in the vertical influences the

radiative forcing per BC mass. Our results highlight the influences of cloud microphysical processes on

BC concentrations and radiative forcing.

## 1 Introduction

Black carbon (BC) is a light-absorbing carbonaceous aerosol resulting from combustion of fossil fuels

or biomass. BC is an important air pollutant that leads to visibility reduction and human health risk. BC

also affects the energy balance of the atmosphere by absorbing solar radiation and interaction with clouds

(Zuberi et al., 2005;Ramanathan and Carmichael, 2008;Bond et al., 2013). In addition, BC deposited in

the Arctic reduces the reflectance of ice and snow, increases the absorption of solar radiation of the

surface, and thus can lead to snow melt. The top of atmosphere direct radiative forcing (DRF) due to all

BC sources was estimated by Bond (2013) to be +0.88 W m$^{-2}$ with 90% uncertainty range of +0.17 W m$^{-2}$ to +1.48 W m$^{-2}$, which is second only to the DRF of $CO_2$. However, after taking aircraft observations

into account,   Wang et al. (2014) and Samset et al. (2014) suggest a weaker global DRF of 0.17- 0.19

W m$^{-2}$ . The large disagreements among models can be mainly attributed to model uncertainties in

simulating BC concentrations (particularly in remote regions). Compared to observations, models

underestimate BC concentrations in the Arctic during winter and spring, while overestimating BC

concentrations in pacific tropical regions (Liu et al., 2011;Winiger et al., 2017;Schwarz et al., 2013).

Moreover, models fail to capture BC vertical profiles measured by aircraft, with overestimates of BC in

the upper troposphere (Schwarz et al., 2013;Schwarz et al., 2010;Schwarz et al., 2017). The inter-model

discrepancies and disagreement between models and measurements reflect uncertainties in emissions,



transport, aging, and dry and wet scavenging of BC simulation. In the remote troposphere, wet scavenging is considered as a primary source of BC simulation uncertainties (Liu et al., 2011;Koch et al., 2009;Schwarz et al., 2010;Croft et al., 2010).

Fresh BC particles are emitted mostly as ultrafine (diameter <100 nm) hydrophobic aerosols and become larger (diameter > 100 nm) hydrophilic particles through the so-called "aging" process, where soluble materials coat BC. During transport, BC can be removed by stratiform cloud (i.e. liquid clouds, mixed-phase clouds and ice-phase clouds) and convective cloud precipitation as in-cloud scavenging and below cloud scavenging. Hydrophilic BC particles are able to act as cloud condensation nuclei (CCN) that lead to liquid stratiform cloud formation (Croft et al., 2005). Through accretion and autoconversion, cloud droplets grow until they are large enough to precipitate. Most BC particles are removed from the atmosphere via precipitation, while a small fraction of BC goes back into an interstitial state during the falling of rain droplets. BC also undergoes mixed-phase and ice cloud scavenging. Primary ice crystals are produced by heterogeneous nucleation and homogeneous nucleation, while collision of ice crystals with supercooled cloud droplets (riming) forms secondary ice crystals. Heterogeneous nucleation can be initiated on (1) ice nuclei (IN) immersed in a cloud droplet (immersion mode), (2) IN in contact with a supercooled cloud droplet (contact mode), and (3) directly on bare IN (deposition mode). The relative importance of the three pathways depends on ambient temperature, water vapour saturation, and properties of ice nuclei. Generally, at temperatures lower than 237 K, deposition freezing and homogeneous freezing dominate, while at temperatures between 237 K and 243 K ice nucleation mainly occurs via contact and immersion freezing. Modelling studies show that if BC is an efficient IN, its impact on cirrus cloud formation would be significant (Penner et al., 2009;Barahona, 2012). Although studies disagree on whether BC can act as IN (Gorbunov et al., 2001;Friedman et al., 2011;Kireeva et al., 2009;Fornea et al., 2009;Dymarska et al., 2006), the majority of laboratory studies argue that BC is a poor IN compared to mineral dust and biological particles, in that BC needs colder temperatures to initiate ice formation (Hoose and Möhler, 2012).

In mixed-phase clouds, observations have found that riming increase BC scavenging efficiency because precipitated ice crystals collect BC in the supercooled droplets of clouds at lower altitudes (Hegg et al., 2011). In addition to the riming, ice crystals can also grow through the Bergeron process—when





water vapour pressure is supersaturated with respect to ice and undersaturated to liquid water, cloud droplets evaporate and condense onto ice crystals/snow. Unlike riming, the Bergeron process decreases BC scavenging efficiency, as cloud-borne BC go back into the interstitial phase. It can explain field measurements at Jungfraujoch in Switzerland where the scavenging fraction of BC decreases by 50%-

55% when ice mass fraction increases from 0 to 0.2 (Cozic et al., 2007). Modelling studies suggest that the Bergeron process is important to the simulation of BC in the Arctic (Fan et al., 2012). Qi et al. (2017) found that the relative importance of the riming and the Bergeron processes in mixed-phase clouds depends on location.

The contribution of convective cloud wet removal to total wet deposition of BC in models ranges

between 10% to 90%, depending on the convective scheme (Textor et al., 2006). Using convection schemes that generate greater convection mass flux and precipitation in atmospheric models tend to predict higher aerosol vertical dispersivity   (Park and Allen, 2015;Allen and Landuyt, 2014).

Many global climate models and chemical transport models employ simplified parameterizations to compute aerosol first-order wet removal rates, based on stratiform and convective cloud fraction,

precipitation production rate, and a solubility factor. The solubility factor represents the fraction of aerosols in cloud droplets multiplied by a tuning factor, and is often fixed in models. A few advanced global climate models (e.g., CAM5, ECHAM5-HAM, HadGEM2-A) explicitly calculate the fraction of aerosols that act as CCN and can subsequently be removed by precipitation. However, even in these advanced models, other cloud processes (e.g., the Bergeron process, riming, cloud water/ice conversion)

only affect cloud microphysics but not in-cloud aerosol concentrations. Thus, most global models treat BC wet scavenging in an incomplete way (Textor et al., 2006;Wang et al., 2011;Croft et al., 2010;Qi et al., 2017). Previous studies suggest that more physically-based schemes in many cases can reduce the disagreement between simulations and observations and highlight the importance of cloud processes in aerosol removal (Vignati et al., 2010;Kipling et al., 2013). Therefore, models that couple aerosol

chemistry with cloud microphysics are essential for accurately simulating BC wet removal and concentrations. Meanwhile, the extent to which different cloud processes can affect BC spatiotemporal distributions still remains uncertain due to a lack of both observations and modelling studies. To our


knowledge, previous studies have never systematically investigated and quantified the effect of each cloud process on BC distributions.

In this study, we aim to improve the simulation of BC wet removal and assess the influence of the aforementioned cloud processes on BC concentrations and radiative forcing. We develop an improved

wet removal scheme that enables BC particles to evolve following cloud processes in a state-of-science earth system model. We quantify the conversion of BC among interstitial, in-cloud-water, in-cloud-ice, in-rain, and in-snow states via different cloud processes. We also perform a series of sensitivity simulations, and investigate the influence of each cloud process on BC concentration distributions and radiative forcing effects.

## 10   2 Methods

### 2.1 Model configuration

Simulations are performed using the state-of-the-science fully coupled Community Earth System Model (CESM) version 1.2.2 (http://www.cesm.ucar.edu/models/cesm1.2.2/), which consists of the Community Atmosphere Model version 5 (CAM5), Community Land model version 4 (CLM4), and

prescribed sea ice and sea surface temperatures (Hurrell et al., 2013). A finite volume dynamical core is employed at 1.9°×2.5° horizontal resolution with 56 levels in the vertical. We nudge the model to GEOS5 offline meteorology (e.g., temperature and wind). Model simulations are performed from 1 January 2008 to 1 August 2011 with the first year discarded as spin-up. The stratiform cloud microphysics scheme used in CAM5 is double moment (Morrison and Gettelman, 2008), predicting

number concentrations and mixing ratios of cloud particles as well as diagnosing number concentrations and mass of precipitation. Cloud microphysical processes include nucleation of cloud droplets, primary ice nucleation, vapour deposition onto cloud ice, evaporation/sublimation of cloud liquid and ice, conversion of cloud liquid to rain by autoconversion and accretion, conversion of cloud ice to snow by autoconversion and accretion, accretion of cloud liquid by snow, self-collection of snow,

self-collection of rain, collection of rain by snow, freezing of cloud liquid and rain, melting of cloud ice and snow, evaporation/sublimation of precipitation, sedimentation of cloud liquid and cloud ice, and convective detrainment of cloud liquid and cloud ice (Gettelman et al., 2008). Parameterization of ice





nucleation for both cirrus clouds and mixed-phased clouds, which predicts ice crystal number

concentrations and calculates ice supersaturation, is based on Liu and Penner (2005) and Liu et al.

(2007) and later updated by Gettelman et al. (2010). Shallow convection is treated with a

parameterization developed by Park and Bretherton (2009) that computes vertical velocity and

fractional area of convection, and more accurately simulates spatial distribution of shallow column

activity. The deep convection scheme in CAM5 is from Zhang and McFarlane (1995). The impact of

aerosols on convective clouds is not considered in the convective cloud parameterizations.

CAM5 couples with seven internal-mixed log-normal aerosol modes (MAM-7), which divide aerosols

into seven modes and predict both mass mixing ratios and number concentrations of aerosol species

(Liu and Ghan, 2010). In order to estimate the influence of cloud processes on BC concentrations, we

add bulk BC tracers to track the conversion of BC in cloud processes, as described in section 2.2. We

use the MACCity emission inventory, which was developed for MACC and CityZen projects

(Lamarque et al., 2010), for anthropogenic BC emissions, and Global Fire Emissions Database (GFED)

version 3 monthly emissions for BC from biomass burning (van der Werf et al., 2010). BC tracers are

unable to affect cloud physics (e.g., cloud droplets and ice crystals formation) and atmospheric physics.

To estimate the direct radiative forcing of BC, we use the Parallel Offline Radiative Transfer (PORT), a

stand-alone tool of CESM. PORT is driven by previous model-generated datasets and uses the code of

Rapid Radiative Transfer Method for global climate models (Conley et al., 2013). PORT is able to

calculate a more reasonable radiative forcing than instantaneous radiative forcing, since it considers

stratospheric temperature adjustment with fixed dynamic heating. We run PORT for four months as

spin-up prior to a full-year simulation, and the output time step is every 1.5 days plus 1 CAM5 time

step. In each output time step, the radiation scheme is called twice with and without the presence of

BC. The difference in net radiation flux at the tropopause between the presence and absence of BC

aerosols is defined as radiative forcing.

**2.2 Wet removal parameterization of BC**

In order to improve model simulations of BC and evaluate the influence of different cloud processes on

BC, we have introduced a new parameterization that explicitly describes BC wet removal. BC particles

are tagged using four BC tracers, hydrophobic BC in the interstitial phase ($BC_{phobic}$), hydrophilic BC





in the interstitial phase ($BC_{philic}$), BC in cloud water ($BC_{water}$), and BC in cloud ice ($BC_{ice}$). These four

BC tracers undergo the same atmospheric processes (except for wet removal processes) as untagged BC.

In order to better calculate wet deposition and the amount of BC returning to the atmosphere during the

evaporation of precipitation, we introduce two diagnostic variables $BC_{rain}$ and $BC_{snow}$ for BC in rain

and snow, respectively. BC conversion among different phases associated with cloud processes are

numerous and usually occur simultaneously. Therefore, instead of modifying the original wet removal

scheme, we add chemical reactions in a pre-processor file to represent BC conversion among different

states due to most cloud processes, except for below cloud scavenging and precipitation evaporation.

These two processes are left out because characterizing them requires column integrated precipitation

calculated in the wet removal module. BC aerosols emit in combination of 80% hydrophobic $BC_{phobic}$

and 20% hydrophilic $BC_{philic}$. A fixed e-folding aging time (36 hours) is assumed to convert $BC_{phobic}$

to $BC_{philic}$. In our study the activation rate is diagnosed from the cloud droplet number concentration

(CDNC) tendency (# $kg^{-1}$ $s^{-1}$) calculated in the cloud microphysics scheme. In the standard CAM5 cloud

microphysics scheme, BC does not serve as IN in ice nucleation (Gettelman et al., 2010). Only sulphate

and dust initiate homogeneous freezing and heterogeneous ice nucleation, respectively. In our study, BC

can serve as IN with the same properties as dust for immersion nucleation, following CAM3 and

ECHAM5-HAM (Liu et al., 2007;Liu and Penner, 2005;Kärcher and Lohmann, 2002). We turn off BC

ice nucleation in one of the sensitivity simulations described in Section 2.3. BC ice nucleation rate is

diagnosed from immersed ice cloud number concentration (ICNC) tendency (# $kg^{-1}$ $s^{-1}$). The rates ($s^{-1}$) of

BC cloud activation $k_{philic \rightarrow water}$ that converts $BC_{philic}$ to $BC_{water}$ and BC ice nucleation

$k_{philic \rightarrow ice}$ that converts $BC_{philic}$ to $BC_{ice}$ are given by

$$k_{philic \rightarrow water} = \frac{CDNC}{N_{aerosol-CCN}} \qquad (1)$$

$$k_{philic \rightarrow ice} = \frac{ICNC}{N_{aerosol-IN}} \qquad (2)$$

where $N_{aerosol-CCN}$ ($N_{aerosol-IN}$) is aerosol number concentration (# $kg^{-1}$) that can serve as CCN (IN).

BC in cloud water can transform into cloud ice through immersion, contact freezing and homogeneous

freezing as well as riming splintering when temperature is low. In turn, BC in cloud ice goes back into


cloud water through melting. The conversion rates of $BC_{water}$ to $BC_{ice}$ ($BC_{ice}$ to $BC_{water}$), $k_{water \rightarrow ice}$ ($k_{ice \rightarrow water}$) are calculated as

$$k_{water \rightarrow ice} \quad (3)$$
$$= \frac{CONTACT + IMMERSION + HOMO + SPLINTERING}{Q_{liq}}$$

$$k_{ice \rightarrow water} = \frac{MELT}{Q_{ice}} \quad (4)$$

where $CONTACT$, $IMMERSION$, $HOMO$, $PLINTERING$ and $MELT$ represent mass mixing ratio tendency (kg kg$^{-1}$ s$^{-1}$) of contact freezing, immersion freezing, homogeneous freezing, riming splintering and melting, respectively. $Q_{liq}$ is grid-average cloud water mixing ratio (kg kg$^{-1}$), $Q_{ice}$ represents grid-average cloud ice mixing ratio (kg kg$^{-1}$).

There are several mechanisms that enable BC in cloud water (ice) to evaporate back into the interstitial

state: evaporation of the cloud, the Bergeron processes, and evaporation (sublimation) of sedimented cloud water (ice) from the upper level to the given level. Rates of $k_{water \rightarrow philic}$ ($k_{ice \rightarrow philic}$) from $BC_{water}$ ($BC_{ice}$) to $BC_{philic}$ can be expressed as

$$k_{water \rightarrow philic} = \frac{EVP\_CLOUD + BERG + EVP\_CSEDI}{Q_{liq}} \quad (5)$$

$$k_{ice \rightarrow philic} = \frac{EVP\_ISEDI}{Q_{ice}} \quad (6)$$

where $EVP\_CLOUD$, $BERG$ and $EVP\_CSEDI$, $EVP\_ISEDI$ represent mass mixing ratio conversion

tendency (kg kg$^{-1}$ s$^{-1}$) from cloud water to water vapour by evaporation of the cloud, the Bergeron process and evaporation of cloud water sedimentation, and sublimation of cloud ice sedimentation, respectively. Autoconversion (i.e., collision and coalescence of cloud droplets to form raindrops) combined with accretion of cloud water by rain converts $BC_{water}$ to $BC_{rain}$ in large rain droplets; $BC_{rain}$ is then removed from the atmosphere. Similarly, snow growth results from collision and coalescence of ice

crystals along with riming (i.e., accretion of cloud water by large ice particles), which can transfer $BC_{ice}$ and $BC_{water}$ to $BC_{snow}$; $BC_{snow}$ is then removed from the atmosphere. The above processes can be expressed as




$$k_{water \to rain} = \frac{PRAO+PRCO}{Q_{liq}} \tag{7}$$

$$k_{ice \to snow} = \frac{PRAIO+PRCIO}{Q_{liq}} \tag{8}$$

$$k_{water \to snow} = \frac{RIMING}{Q_{liq}} \tag{9}$$

where $k_{water \to rain}$ ($k_{ice \to snow}$) is conversion rate (s$^{-1}$) from $BC_{water}$ ($BC_{ice}$) to BC in rain droplets (ice crystals), and $k_{water \to snow}$ is reaction rate (s$^{-1}$) from $BC_{water}$ to accretion on ice particles. PRAO (PRAIO) is accretion rate (kg kg$^{-1}$ s$^{-1}$) of cloud water (ice) by rain (snow), PRCO (PRCIO) is

5  autoconversion rate (kg kg$^{-1}$ s$^{-1}$) of cloud water (ice), and RIMING represents cloud water mixing ratio

tendency of riming (kg kg$^{-1}$ s$^{-1}$).

When it comes to convection scavenging, unlike large scale precipitation, we assume that BC can be

totally removed in a column over a sub grid box where convection precipitation occurs. Deposition rates

$k_{phobic \to convection}$ ($k_{philic \to convection}$) of $BC_{phobic}$ ($BC_{philic}$) by precipitation can be represented as

$$k_{phobic \to convection} = \frac{RRDP+RRSH}{Q_{liq}+Q_{ice}} \tag{10}$$

$$k_{philic \to convection} = \frac{RRDP+RRSH}{Q_{liq}+Q_{ice}} \tag{11}$$

where RRDP is deep convection precipitation production rate (kg kg$^{-1}$ s$^{-1}$), and RRSH is shallow

convection precipitation production rate (kg kg$^{-1}$ s$^{-1}$).

### 2.3 Sensitivity simulations

The simulation using CESM with our improved wet removal parameterization is defined as BASE. Eight

15  sensitivity simulations are conducted to investigate the spatiotemporal distributions of BC responses to

eight cloud processes. These eight processes are more important than other cloud processes as reported

in section 3. We turn off the impact of each cloud process on BC in each sensitivity simulation, including

no convective scavenging (NO CONVECTION), cloud activation (NO CCN), ice nucleation (NO IN),

riming (NO RIMING), below cloud scavenging (NO BELOW CLOUD), the Bergeron processes (NO

20  BERGERON), evaporation/sublimation of sedimented cloud liquid and cloud ice (NO CLOUD EVAP),

and evaporation/sublimation of precipitation (NO PRECIP EVAP); these processes are rarely fully



considered in bulk BC aerosol models. The fractional changes in BC concentrations relative to BASE are calculated to quantify the influence of each cloud process on BC. Note that there is no radiative feedback on the climate system from bulk BC tracers in this study. Therefore, changes in aerosol concentrations do not impact climate in these simulations.

5 **2.4 Model evaluation**

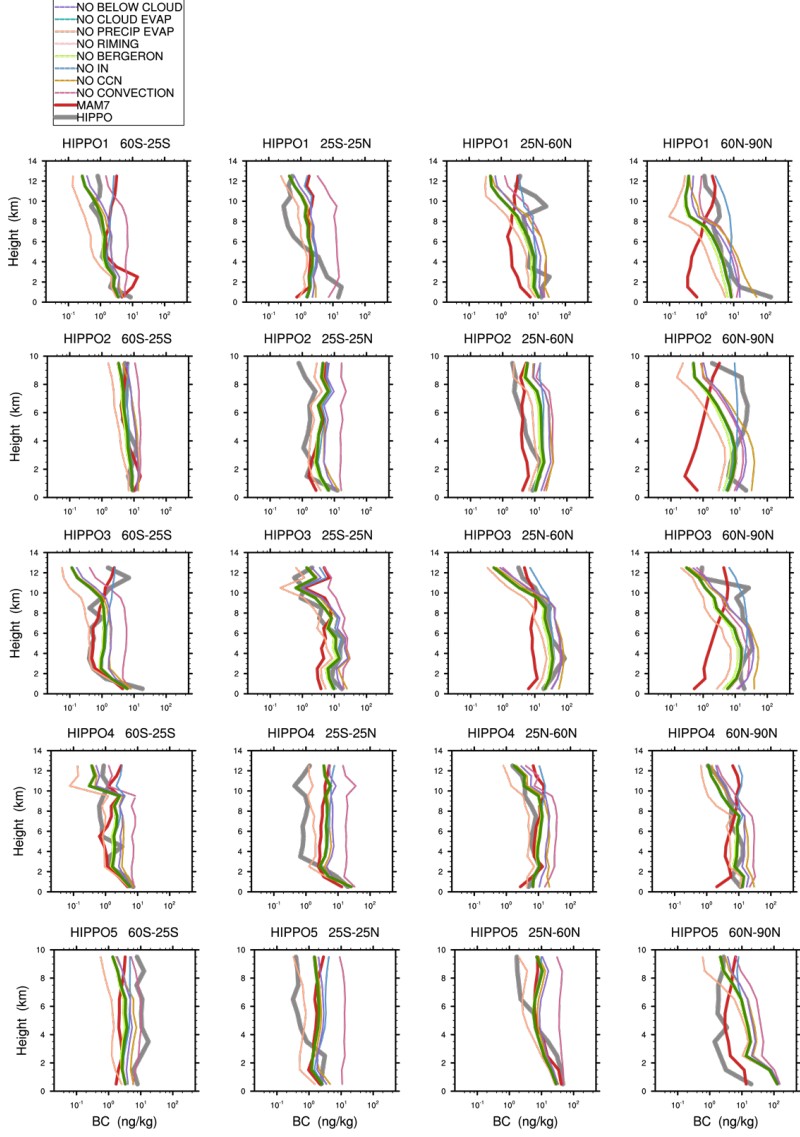





**Figure 1. Vertical profiles of observed and simulated BC concentrations over 1 km altitude bins along the flight tracks of HIPPO 1-5, averaged over 60–20° S, 20° S–20° N, 20–60° N, and 60–90° N. Solid grey thick line, solid red thick line, solid green thick line represent values from HIPPO observations, default model with**

**MAM7 aerosol scheme, and the improved model using our wet removal scheme described in section 2.2 (BASE), respectively. Thin lines represent the vertical profiles of CESM sensitivity simulations when the influence of one cloud process on BC is turned off. The sensitivity simulations are described in section 2.3, including NO CONVECTION (no convection scavenging), NO CCN (no cloud activation), NO IN (no ice nucleation), NO RIMING (no riming), NO BELOW CLOUD (no below cloud scavenging), NO BEGERON**

**(no Bergeron process), NO CLOUD EVAP (no evaporation of cloud water/ice sedimentation), NO PRECIP EVAP (no evaporation of rain/snow).**

In order to evaluate our new parameterization, we compare model simulation results with aircraft measurements from HIAPER Pole-to-Pole Observation (HIPPO). Five HIPPO campaigns carry a single-

particle soot photometer (SP2) to measure BC concentrations over the remote pacific spanning from 85°N to 67°S. Because the aircraft both ascends and descends along each flight track, HIPPO generates vertical profiles of BC concentrations. Compared to the default MAM7 scheme, the vertical profiles of BC simulated using our improved wet removal parameterization are much closer to the HIPPO1-4 observations (Fig. 1). In particular, BC vertical profiles simulated by our improved model fit well with

HIPPO1-5 over high-latitudes in the Northern Hemisphere (NH) and the Southern Hemisphere (SH) in both magnitude and pattern.

Here we also use the mean normalized absolute error (MNAE) and mean normalized bias (MNB) as indicators of model performance, since they weigh high and low bias equally (Zhang et al., 2015). MNAE and MNB can be computed as

$$\text{MNB} = \frac{1}{N}\sum \text{nlat} \sum \text{nalt} \frac{1}{2} \frac{(\text{BC}_m(i,j) - \text{BC}_o(i,j))}{(\text{BC}_m(i,j) + \text{BC}_o(i,j))} \tag{12}$$

$$\text{MNAE} = \frac{1}{N}\sum \text{nlat} \sum \text{nalt} \frac{|\text{BC}_m(i,j) - \text{BC}_o(i,j)|}{\text{Min}(\text{BC}_m(i,j), \text{BC}_o(i,j))} \tag{13}$$



where $i$ represents latitude bin indices, $j$ represents altitude bin indices, nalt =10 and nlat = 15 are the total number of altitude bins (every 1 km from 0 to 10 km) and latitude bins (every 10° from 70°S to 80°N), respectively. $N$ = 150 is product of nlat and nalt, representing the total number of latitude and altitude bins. Model and observation results are averaged over latitude and altitude bins. Compared to

the default MAM7 scheme, our improved scheme considerably reduces both MNAE and MNB for HIPPO1-4. In particular, the MNAE of our improved model is a factor of 13 smaller than that of MAM7 for HIPPO1 (in winter).

**Table 1. The mean normalized absolute error (MNAE) and mean normalized bias (MNB) for BC vertical**
**profiles from BASE (using our improved wet scavenging scheme for BC) and the default MAM7 scheme, compared to vertical profiles measured by HIPPO 1-5. MNAE and MNB are defined in section 2.4.**

|        |       | MNAE  | MNB  |
|--------|-------|-------|------|
| HIPPO1 | BASE  | 9.8   | 1.02 |
|        | MAM7  | 127.2 | 1.26 |
| HIPPO2 | BASE  | 5.1   | 0.98 |
|        | MAM7  | 24.4  | 1.31 |
| HIPPO3 | BASE  | 15.2  | 1.25 |
|        | MAM7  | 39.4  | 1.35 |
| HIPPO4 | BASE  | 5.1   | 0.91 |
|        | MAM7  | 5.8   | 1.02 |
| HIPPO5 | BASE  | 5.1   | 0.90 |
|        | MAM7  | 5.0   | 1.0  |

**3 Budget of BC**

Our improved BC wet removal scheme tightly links cloud processes with BC wet removal by considering
BC conversion among six states (i.e., $BC_{phobic}$, $BC_{philic}$, $BC_{water}$, $BC_{ice}$, $BC_{rain}$, $BC_{snow}$) as described




in section 2.2. Therefore, the solubility factor of BC is no longer a constant parameter but spatially and

temporally dynamic. In order to quantitatively investigate BC conversion along with each cloud process,

we calculate the global total annual mean BC conversion rate due to each process (Fig. 2). 80% of BC

(233 kg/s) is emitted as hydrophobic BC and 20% is emitted as hydrophilic (49 kg/s). Global BC aging

rate that converts interstitial $BC_{phobic}$ to $BC_{philic}$ is 181 kg/s.

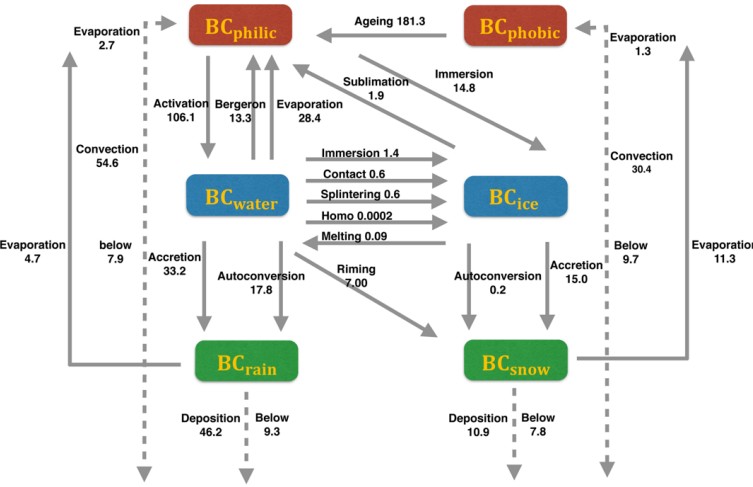

**Figure 2. Global budget of BC conversion (kg/s) among interstitial hydrophobic BC ($BC_{phobic}$), interstitial**
**hydrophilic BC ($BC_{philic}$), BC in cloud water ($BC_{water}$), BC in cloud ice ($BC_{ice}$), BC in rain ($BC_{rain}$), and BC**
**in snow ($BC_{snow}$) due to different cloud processes and aging. The conversion rates shown in the figure**
**represent global total values averaged for year 2009.**

Convection scavenging is computed to be the one of the most influential factors to both $BC_{phobic}$

and $BC_{philic}$ simulations in this study. The rates of convection scavenging are 55 kg/s and 30 kg/s for

$BC_{phobic}$ and $BC_{philic}$, respectively. The rate of total BC removal via convection scavenging (85 kg/s) is

slightly lower than the rate of activation processes (106 kg/s). BC can also be removed with stratiform

precipitation from liquid clouds and ice clouds. Liquid cloud scavenging starts with BC activation (106

kg/s), whose rate is an order of magnitude greater than the rate of BC ice nucleation. In cold clouds, BC

may immerse into water droplets and the global total conversion rate of BC ice nucleation is 15 kg/s.

The Bergeron process refers to the mechanism that allows ice crystals to grow at the expense of cloud

water evaporation. This process releases BC in cloud water to the interstitial state in the atmosphere





($BC_{philic}$) at a conversion rate of 13 kg/s globally, one order of magnitude smaller than BC activation. Other cloud evaporation processes that convert cloud water to water vapour (28 kg/s in total) can also release BC in cloud droplets, converting $BC_{water}$ to $BC_{philic}$. The conversion can occur via (1) cloud evaporation and regeneration within a model time step, and (2) evaporation during cloud water

5    sedimentation from a model layer above. Similarly, sublimation of ice crystal sedimentation from the upper level converts $BC_{ice}$ to $BC_{philic}$, at rate of 15 kg/s.

When temperatures are below freezing, cloud water becomes supercooled. BC within supercooled droplets can transform into ice crystals through four processes: immersion freezing, contact freezing, rime-splintering, and homogeneous freezing. Their conversion rates are less than 1.5 kg/s globally,

10    smaller than most cloud processes. Conversion of $BC_{ice}$ to $BC_{water}$ through melting is the slowest (0.085 kg/s) among all cloud processes.

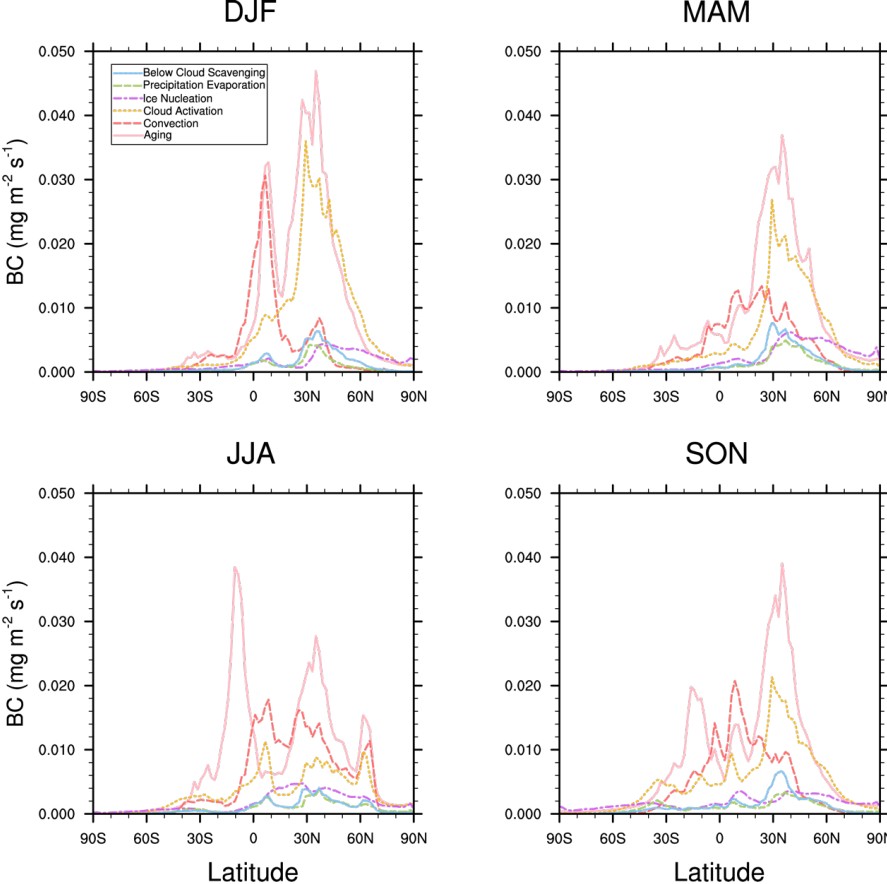



**Figure 3. Zonal mean column total BC conversion rates (mg m$^{-2}$ s$^{-1}$) due to processes related to cloud formation and precipitation including below cloud scavenging, precipitation evaporation, ice nucleation, cloud activation, convective scavenging, and aging during four seasons (DJF, MAM, JJA and SON) of year 2009.**

The rest of cloud water (cloud ice) turns into large rain droplets (snow) through accretion and autoconversion, with conversion rates at 51 (15) kg/s. In addition, riming, another important mechanism of ice growth converts $BC_{water}$ to $BC_{snow}$ at the rate of 7 kg/s, about half the increase in interstitial BC due to the Bergeron process. The majority of BC in clouds is removed from the atmosphere via rain and snow. However, when rain (snow) evaporates (sublimates), 10% (75%) of $BC_{water}$ ($BC_{ice}$) is released

back into atmosphere. Below cloud scavenging washes out BC in interstitial phase. Washout rates of BC from convective and large scale stratiform precipitation are roughly the same; the total below-cloud BC scavenging rate for all clouds is 35 kg/s globally.

     Figures 3 and 4 show the zonal mean column total BC conversion rates due to the aforementioned processes over four seasons (i.e. winter (DJF), spring (MAM), summer (JJA) and autumn (SON)). Here

we define seasons based on NH. We divide cloud processes into two groups: in-cloud processes (fig. 4) and other processes related to cloud formation and precipitation (fig. 3). Processes in the latter group are an order of magnitude larger than former ones. BC conversion rates among different cloud processes show large spatial and seasonal variations as they strongly depend on background aerosol concentrations, temperature, humidity, and other meteorological factors.

Figure 3 shows that BC removal from convective scavenging peaks at around 0° in winter, while values are greater in NH mid-latitudes in summer with smaller zonal variation. BC conversion rate for cloud activation is largest in winter in the NH, reaching its maximum at 30°N, consistent with the peak of aging rate. This is because higher average emissions around 30°N lead to higher conversion rates of $BC_{phobic}$ to $BC_{philic}$, which can then act as CCN. The conversion of $BC_{philic}$ to $BC_{ice}$ through ice

nucleation is a dominant process in high-latitude regions, because of low ambient temperatures. Conversion rates of BC due to evaporation during precipitation and below cloud scavenging are relatively small compared to other processes and show similar zonal and seasonal variations.



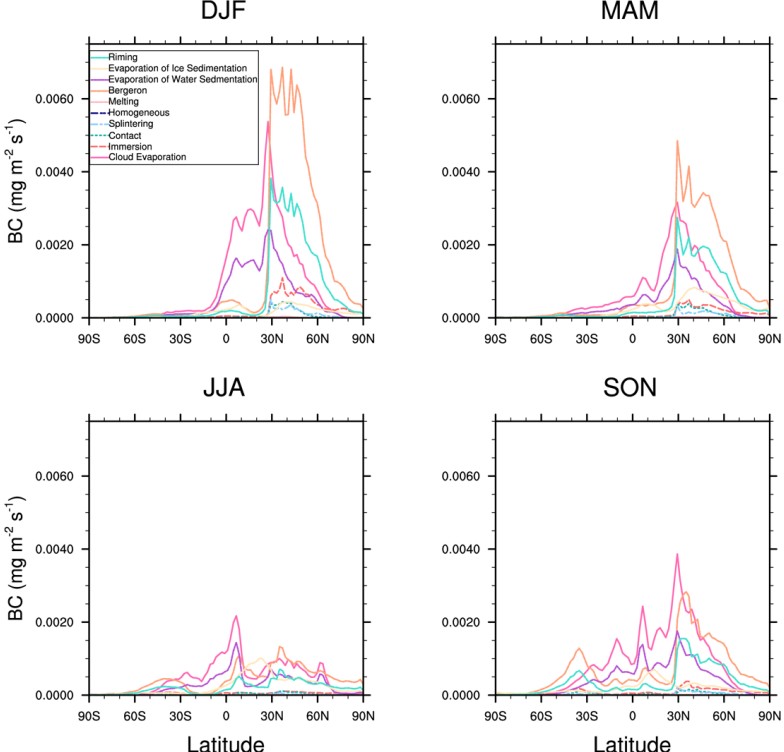

**Figure 4. Same as Figure 3, but for in-cloud processes including evaporation of cloud, immersion freezing, contact freezing, rime splintering, homogeneous freezing, melting, Bergeron process, evaporation of cloud water sedimentation, evaporation of cloud ice sedimentation, and riming.**

The in-cloud processes show distinct seasonal variations in altering BC conversion rates (Fig. 4). Based on the patterns in zonal mean column BC conversion rates, the processes can be grouped into (1) processes related to cloud water and ice formation and conversion and (2) processes related to evaporation. In order to explain the patterns, zonal mean column of cloud water conversion rates during

10 several cloud processes are plotted in Fig. S1. BC transformation among cloud water and ice (e.g., the Bergeron process, riming, heterogeneous freezing, homogeneous freezing, splintering, melting) show seasonable features including (a) significant higher BC conversion rates during winter in 30°N-60°N (fig. 4), due to higher cloud water conversion rates for riming and the Bergeron processes in the mid-latitudes




(Fig. S1), and (b) comparatively uniform conversion rate for summer from 60°S to 60°N (Fig. 4), because

the Bergeron process and riming cloud water conversion rates have less zonal variations (Fig. S1) in

summer. Unlike BC conversion during cloud water and ice transformation, the conversion rate due to

evaporation of cloud and cloud water sedimentation peaks in tropical regions and decreases with

5      increasing latitude. This is because cloud water evaporation peaks in the tropics (fig. s1). Figure 4 also

clearly indicates that the BC conversion rates related to cloud processes are much greater in the NH than

the SH, especially during winter.

## 4 The influence of cloud processes on BC spatial and vertical distributions

### 4.1 BC spatial distribution influenced by individual cloud processes

10     As described in Section 2.3, we perform sensitivity simulations to investigate the influence of eight cloud

processes on spatiotemporal distributions of BC.

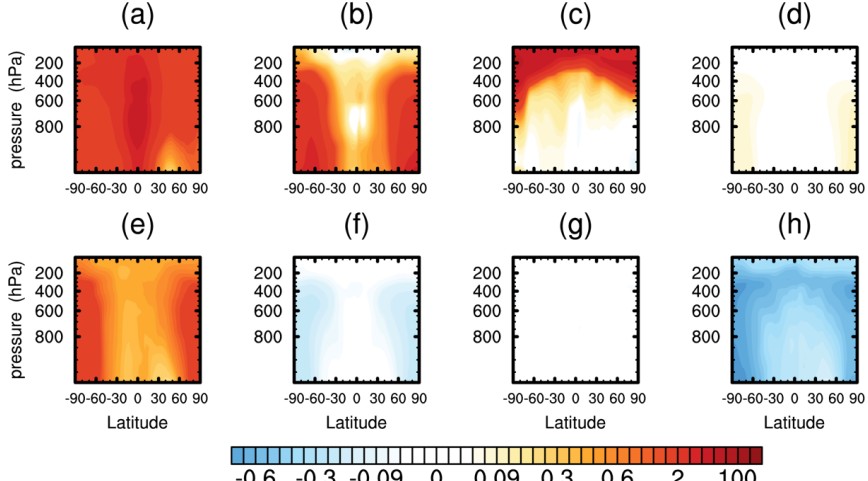

**Figure 5. Annual zonal mean fractional changes (unitless) for year 2009 in BC concentrations relative to BASE
in eight sensitivity simulations when the influence of one cloud process on BC is turned off. The sensitivity
simulations are described in section 2.3, including (a) NO CONVECTION (no convection scavenging), (b) NO
CCN (no cloud activation), (c) NO IN (no ice nucleation), (d) NO RIMING (no riming), (e) NO BELOW
CLOUD (no below cloud scavenging), (f) NO BERGERON (no Bergeron process), (g) NO CLOUD EVAP (no
evaporation of cloud water/ice sedimentation), and (h) NO PRECIP EVAP (no evaporation of rain/snow).**



Figure 5 presents vertical distributions of zonal mean BC concentrations when each cloud process is turned off. Turning off convection scavenging results in considerable increases in BC concentrations, especially over the tropics where convection is prevalent (see Fig. 5a). Our results are supported by Lund et al. (2017), who found convective scavenging to be a key parameter in determining the BC

concentration in OsloCTM2-M7. The cloud activation is another important controller of BC concentrations. This is because activation determines whether BC can be removed by stratiform liquid cloud wet scavenging (see Fig. 5b). Absolute changes in BC concentrations induced by turning off cloud activation are generally larger in the lower troposphere over the North Pole (Fig. S3b), while the fractional increases in BC concentrations are more evident in the free troposphere at high latitude in both

NH and SH (Fig. 5b). This is because during long-distance transport, longer BC life time allows more BC to reach the remote atmosphere where baseline BC concentrations are comparatively low (Wang et al., 2014), and thus the fractional change increases drastically. The seasonal variation is also distinctive; both fractional differences and absolute differences over NH induced by turning off CCN activation is higher in winter (Fig. S3). Similarly, BC fractional increases due to turning off below cloud scavenging

are larger over the North and South Poles (Fig. 5e). However, the absolute increases in BC when turning off below cloud scavenging reach a maximum at mid-latitudes near surface (Fig. S3e).

Wet removal by ice clouds is another important process that decreases BC lifetime in the atmosphere. It takes place where mixed-phase and cold clouds occur. As a result, greater BC burden increases for NO IN relative to BASE are found at high altitudes over tropical and high-latitude regions (Fig. 5(c)). The

absolute differences between NO IN and BASE simulations show distinctive seasonal variations, with much larger increases over high-latitudes in Northern Hemisphere during winter than summer (Fig. S3c)

Similarly, riming shortens BC life time but its effect is weaker than aforementioned processes (Fig. 5). As shown in figure 5d, the influence of riming is more important over mid- and high-latitudes where mix-phase clouds are prevalent. The influence of riming on BC is less important than ice nucleation,

because even if BC in supercooled cloud water is not collected by ice, it would still eventually be removed by rain droplets. Fan et al. (2012) highlight the importance of riming in increasing scavenging efficiency in mixed-phase clouds. However, in their study, BC scavenging by ice and snow happens by implicitly describing the riming and homogeneous freezing, so their "riming" essentially represents the total effect





of these two processes. On the contrary, in our study, we explicitly track BC ice cloud scavenging due to immersion freezing and riming separately. Therefore, the fractional change due to riming in the model is not as significant as reported in Fan et al. (2012).

The remaining processes (the Bergeron process, evaporation of cloud sedimentation, and evaporation of precipitation) have the opposite effect on BC distributions. The Bergeron process releases BC in cloud water, enhancing BC long-range transport and therefore increasing BC concentrations at high altitudes in the Arctic. Therefore, NO BEGERON decreases BC concentrations in the Arctic relative to BASE (Fig. 5(f)). Figure S2 shows that in winter (summer), fractional increases of BC in NO BERGERON are the strongest in the North Pole due to low baseline BC concentrations and the prevalence of mixed-

phased clouds. The fractional changes relative to BASE for NO BERGERON and NO RIMING are similar in pattern and magnitude but opposite in sign (Fig. 5d,f). The only exception is that over the tropics, the Bergeron (riming) process leads to greater fractional changes in BC at higher (lower) altitudes, consistent with the higher tendency of the Bergeron (riming) processes (Fig. S5).

Another process that enables in-cloud $BC_{water}$ and $BC_{ice}$ to return to interstitial BC in the atmosphere

is evaporation of cloud water/ice sedimentation. In our sensitivity simulation, we turn off both evaporation and sublimation of cloud water and ice sedimentation. Unlike the Bergeron process, the fractional decreases in BC of NO CLOUD EVAP are more significant at lower altitudes (fig. S3g) due to more intense evaporation of cloud sedimentation (fig. S5d). Absolute changes in BC concentrations are larger at lower latitudes (Fig. S3g). On the other hand, evaporation of rain and sublimation of snow nearly

uniformly increases BC concentrations at all altitudes over NH (Fig. S3h).

We find that the because of low background concentrations in BASE, influences of cloud processes on BC concentrations are larger over polar regions, particularly during winter. In addition, BC concentrations over polar regions are challenging for model simulation due to the uncertainties in BC long range transport (e.g., models usually underestimates BC concentration over polar regions in NH

winter and spring (Liu et al, 2011) (Liu et al., 2011)) . Our results highlight the importance of properly characterizing the influence of cloud and wet scavenging processes on BC at high-latitudes.

**4.2 The influence of cloud processes on vertical distribution of BC globally and over the Pacific.**



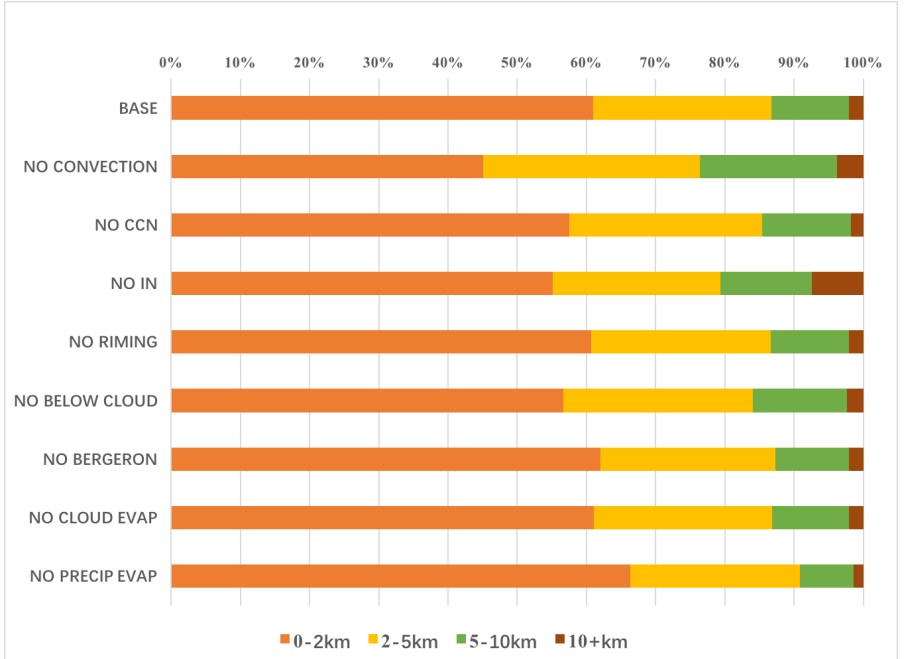

**Figure 6. Fraction of global BC burden at four altitude bands for year 2009 in the BASE simulation using our new wet removal scheme described in section 2.2, and sensitivity simulations when the influence of each cloud process on BC is turned off. The sensitivity simulations are described in section 2.3, including NO CONVECTION (no convection scavenging), NO CCN (no cloud activation), NO IN (no ice nucleation), NO RIMING (no riming), NO BELOW CLOUD (no below cloud scavenging), NO BERGERON (no Bergeron process), NO CLOUD EVAP (no evaporation of cloud water/ice sedimentation), NO PRECIP EVAP (no evaporation of rain/snow).**

In order to demonstrate the influence of cloud processes on vertical distributions of BC concentrations, we calculate the fraction of global total BC burden over four altitude bands in BASE and eight sensitivity simulations (Fig. 6). For BASE, BC burden below 2 km takes the largest fraction (61%) of total BC burden; the fraction decreases with increasing height of altitude bands. NO CONVECTION induces the most significant change to the vertical distribution of BC mass; the fraction of BC at 2-5, 5-10 and above 10 km increases, while that below 5 km decreases. For NO IN, BC cannot serve as ice nuclei, and therefore wet scavenging in ice clouds at high altitudes decreases, leading to large increases in the fraction of BC burden above 10 km. Considering the fraction of BC that is above 5 km and below 5km, we find that simulations of NO CONVECTION, NO CCN, NO IN, NO RIMING and NO BELOW CLOUD



increase BC fraction above 5 km and decrease BC fraction below 5 km. In contrast, simulations of NO BERGERON, NO CLOUD EVAP and NO PERCIP EVAP decrease BC fraction above 5 km.

Figure 1 also shows comparisons between sensitivity simulations and BASE for BC measured during HIPPO1-5 aircraft observaitons over the Pacific Ocean in four seasons. The results in Figure 1 are divided

into four latitude bands. Over 60°S-25°S and the tropics, turning off the influence of convection scavenging leads to the largest increases in BC concentrations at all altitudes, particularly in summer. On the contrary, turning off evaporation of precipitation contributes to the largest reductions in BC concentrations. Over 60°N-90°N, the most significant cloud process determining BC vertical profiles along the HIPPO trajectory is BC ice nucleation; the changes in BC concentrations in NO IN relative to

BASE increase with altitude, and the effect is strongest in NH during winter. During HIPPO2-3, NO IN (i.e., the sensitivity simulation where BC cannot act as IN) would better match observed BC at high altitudes over 60°N-90°N. In addition to NO IN, NO CLOUD ACTIVATION and NO BELOW CLOUD can also significantly increase BC concentrations with larger changes near surface. This is because cloud activation is predominantly below 800 hPa (Fig. S5f), thus below cloud scavenging removes more BC at

lower altitudes. In tropical regions, the vertical profiles for NO CLOUD ACTIVATION and NO BELOW CLOUD are similar, while at high latitudes in NH, cloud activation can induce the largest changes in BC concentrations near the surface, as compared to other cloud processes. Excluding the effect of precipitation evaporation (NO PRECIP EVAP) decreases BC concentrations at high altitudes; NO PRECIP EVAP better matches with HIPPO 1-5 observations over 25°S-25°N than BASE. Other cloud

processes have relatively minor influences on BC vertical profiles. For example, including the influence of the Bergeron process in BASE leads to slightly higher BC concentrations below 5 km relative to NO BERGERON; the influence is more significant over 60°N-90°N than other latitudes (Figure 1). Our results suggest that to match HIPPO observations, it is important that atmosphere models accurately simulate how cloud convection scavenging, evaporation of precipitation, cloud activation, ice nucleation,

and the Bergeron process affect BC concentrations.



## 5 Radiative forcing of BC

Table 2 summarizes the global BC burden and corresponding direct radiative forcing (DRF) in our simulations as well as other studies. The global-mean burden is 85 Gg in the BASE simulation, 23% lower than the original aerosol scheme (MAM7). The burden of BC ranges from 73 Gg to 151 Gg across

the sensitivity simulations. The largest increase in BC burden results from removing the effect of convection scavenging, followed by aerosol activation. The largest reduction in BC burden (12 Gg) is from removing the effect of precipitation evaporation.

**Table 2. Black carbon burden and corresponding radiative forcing for year 2009 simulated by the default MAM7 scheme, BASE (with our improved wet removal scheme), and eight sensitivity simulations including NO CONVECTION (no convection scavenging), NO CCN (no cloud activation), NO IN (no ice nucleation), NO RIMING (no riming), NO BELOW CLOUD (no below cloud scavenging), NO BERGERON (no Bergeron process), NO CLOUD EVAP (no evaporation of cloud water/ice sedimentation), and NO PRECIP EVAP (no**

**evaporation of rain/snow). Values are reported for three previous studies as well.**

| Case | Burden (Gg) | Direct radiative forcing (W m$^{-2}$) |
| --- | --- | --- |
| MAM7 | 100 | 0.16 |
| BASE | 85 | 0.13 |
| NO CONVECTION | 151 | 0.33 |
| NO CCN | 106 | 0.23 |
| NO IN | 93 | 0.18 |
| NO RIMING | 85 | 0.13 |
| NO BELOW CLOUD | 103 | 0.19 |
| NO BERGERON | 82 | 0.12 |
| NO CLOUD EVAP | 84 | 0.13 |
| NO PRECIP EVAP | 73 | 0.09 |
| Wang et al (2014) | 77 | 0.19 |
| Schulz et al (2006) | 118 | 0.27 |
| Bond et al (2013) | 282 | 0.65-0.90 |



The global mean direct radiative forcing of BC simulated using our improved wet removal scheme (BASE) is 0.13 W m$^{-2}$, lower than the default MAM7 aerosol scheme (0.16 W m$^{-2}$) and previous studies (Table 1). The Fifth IPCC Assessment Report estimates the DRF of BC to be 0.6 W m$^{-2}$ (Boucher et al., 2013), and Bond et al. (2013) report a slightly higher estimate of 0.71 W m$^{-2}$. Schulz et al. (2006)

suggest a lower DRF of 0.27±0.06 W m$^{-2}$ based on nine AeroCom models. Wang et al (2014) improve wet removal processes to better match HIPPO observations and report a lower DRF (0.19 W m$^{-2}$) than previous studies. Schwarz et al. (2013) indicate that AeroCom models overestimate BC burden when compared with HIPPO observations, especially in the upper troposphere. Our estimated BC DRF is lower than previous studies because of the difference in (a) schemes used to simulate BC distributions

and (b) the tools used to estimate radiative forcing. BC DRF in the sensitivity simulations ranges from 0.09-0.33 W m$^{-2}$. Turning off the influence of convection scavenging, cloud activation, ice nucleation, riming, and below cloud scavenging processes on BC increases BC DRF. On the other hand, a notable decrease in DRF is observed when the evaporation of precipitation is turned off and a slight reduction is observed in NO BERGERON. The influence of cloud water/ice evaporation on DRF is negligible.

Note that the fractional change (increase or decrease) is higher for BC DRF than BC burden. For example, BC DRF in the simulation without cloud activation is 0.23 W m$^{-2}$, 72 % higher than DRF in BASE, while the BC burden in NO CCN is 107 Gg, only 26 % higher than BASE. This is because NO CCN reduces wet scavenging rate of BC, allowing more BC to transport to above 5 km. The direct radiative forcing per BC mass increases with altitude (Samset and Myhre, 2015;Samset et al., 2013).

The same logic applies to other sensitivity simulations. As discussed in section 4.2, simulations with increased BC burden (i.e. NO CONVECTION, NO CCN, NO IN, NO RIMING, NO BELOW CLOUD) show increased BC fraction at high altitudes, while simulations with decreased BC burden (i.e. NO BERGERON, NO CLOUD EVAP, NO PRECIP EVAP) show decreased BC fraction at high altitudes. Our results show that cloud processes can also influence the efficiency of BC acting as a

radiative forcing agent (direct radiative forcing per BC mass) via changing the vertical distribution of BC.





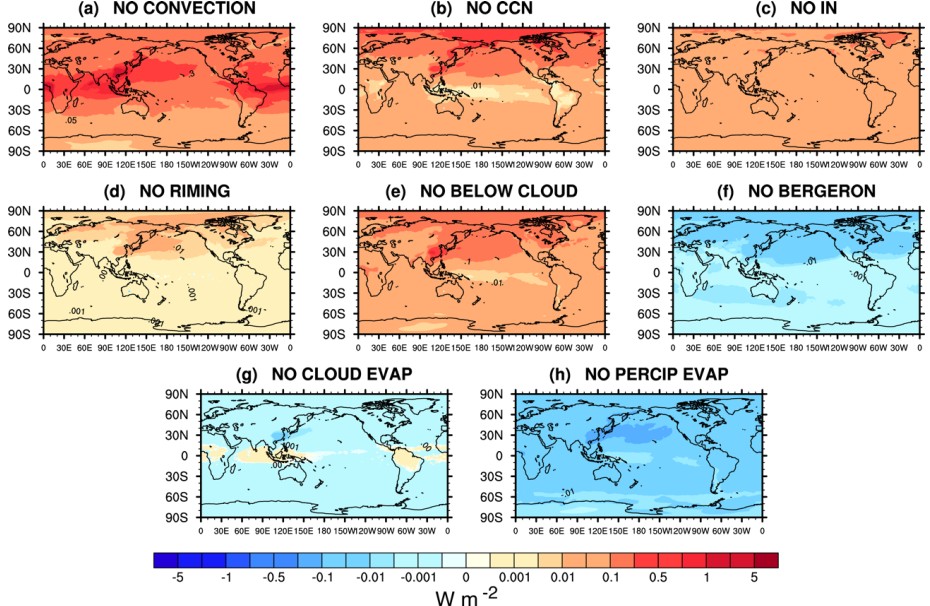

**Figure 7. Change in global radiative forcing of BC estimated by sensitivity simulations relative to BASE for year 2009. The sensitivity simulations are described in section 2.3, including (a) NO CONVECTION (no convection scavenging), (b) NO CCN (no cloud activation), (c) NO IN (no ice nucleation), (d) NO RIMING (no riming), (e) NO BELOW CLOUD (no below cloud scavenging), (f) NO BERGERON (no Bergeron process), (g) NO CLOUD EVAP (no evaporation of cloud water/ice sedimentation), and (h) NO PRECIP EVAP (no evaporation of rain/snow).**

Figure 7 shows the differences in BC DRF between sensitivity simulations and BASE. BC DRF for NO CONVECTION (Fig. 7a) increases, with maximum increases over tropical regions. BC DRF in sensitivity simulations for NO CCN (fig. 7b), NO IN (Fig. 7c), NO RIMING (Fig. 7e) and NO BELOW SCAVENGING (Fig. 7h) increases globally relative to BASE with similar spatial patterns: (a) greater changes in NH than SH, (b) greater changes in mid- and high- latitudes than tropical regions in NH, and (c) maximum increases occurring over East Asia due to higher changes in column burden (Fig. S4). In contrast, NO BERGERON (Fig. 7f), NO CLOUD EVAP (fig. 7g), NO PRECIP EVAP (Fig. 7h) decreases DRF globally relative to BASE. The pattern for DRF decreases is similar with aforementioned pattern for DRF increases, but opposite in sign. The only exception is that DRF decreases due to turning off



precipitation evaporation reaches its maximum over the North Pacific. In general, changes in BC DRF has a similar spatial pattern as changes in BC column burden for all sensitivity simulations (Fig. S4).

Our results indicate that cloud processes and their interactions with aerosols can greatly influence BC DRF, bringing uncertainties in BC radiative forcing estimates. Turning off liquid cloud activation

and convection scavenging in particular can increase BC DRF by about a factor of two. To improve estimates of the climate effects of BC and future climate change (under presumably changing BC emissions), it is critical to properly characterize BC wet removal associated with convective scavenging, cloud activation, ice nucleation, below cloud scavenging, and evaporation of precipitation in global models.

**6 Conclusions**

In this study, we develop a wet removal scheme that explicitly describes the influence of cloud processes on BC in CESM, a global climate model. We add six BC tracers for interstitial hydrophilic BC, interstitial hydrophobic BC, BC in cloud water, BC in cloud ice, BC in rain, and BC in snow; we link the conversion of BC among different phases with cloud microphysical processes. Compared to the original scheme in

CESM (i.e., MAM7), our improved wet scavenging scheme greatly reduces bias against HIPPO 1-4 aircraft observations.

Using the improved wet removal scheme, we calculate global total annual mean BC conversion rates among different phases. We conclude that the eight most important cloud processes that contribute to the largest conversion rates are convection scavenging, cloud activation, ice nucleation, below cloud

scavenging, evaporation of precipitation, riming, the Bergeron process, and evaporation of clouds. The conversion rates of former five processes are almost an order of magnitude higher than latter three processes, while the latter ones show distinct seasonal variations in the Northern Hemisphere with maximum values in winter and minimum values in summer.

To further investigate the influence of the aforementioned eight cloud processes on BC spatiotemporal

distributions, we run eight sensitivity simulations, each of which excludes the influence of one cloud process on BC. BC concentrations at high latitude are found to be more sensitive to most cloud processes





relative to BC at lower latitudes. The only exceptions are convective scavenging and ice nucleation, which mainly influence BC over topical regions and at high altitudes, respectively.

As for BC vertical distributions, turning off the influence of convective cloud scavenging on BC can largely increase the fraction of total column BC above 2 km and decrease that below 2 km. Turning off

the effect of ice nucleation can greatly increase the fraction of BC above 10 km. Turning off the Bergeron process leads to negligible change in the vertical distribution of globally averaged BC but lower BC concentrations at low altitudes over the North Pacific Ocean. We find that sensitivity simulations that lead to higher (lower) BC burden consistently have larger (lower) fraction of total column BC above 5 km.

Our baseline simulation yields a global BC burden of 85 Gg, with corresponding direct radiative forcing (DRF) of 0.13 W m$^{-2}$. Our estimate is lower than previous studies. The BC burden in our sensitivity simulations range from 73 Gg to 105 Gg, with corresponding DRF of 0.09-0.33 W m$^{-2}$. The fractional change in DRF relative to our baseline (BASE) is larger than fractional changes in BC burden for every sensitivity simulation. This is because cloud processes can also influence the direct radiative

forcing efficiency of BC because cloud processes can change BC vertical distributions, and DRF per BC mass increases with altitude.

Our work highlights the importance of cloud processes on BC burden, spatiotemporal distribution, and radiative forcing. In particular, we find that BC is most sensitive to convective scavenging, cloud activation, ice nucleation, below cloud scavenging, evaporation of precipitation, and the Bergeron

process. We suggest that future work prioritize improving representation of these cloud processes on BC in global climate models.





*Data availability.* Data are available upon request from the corresponding authors.

5 *Author contributions.* JX and JL designed the study. JX performed all analysis with help of JZ, KY and XH. JX and JZ prepared the figures and wrote the manuscripts together. YW and SX provided computational support. GBW and ST provided technical guidance.

*Competing interests.* The authors declare that they have no conflict of interest.

*Acknowledgements.* This work was supported by funding from the National Natural Science Foundation of China (under award nos. 41671491, 41571130010, 41390240); the National Key Research and Development Program of China (2016YFC0206202); the 111 Project (B14001); and the National Science
15 Foundation under grants CBET-1512429 and 1752522.





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
