# Peer review of "Influence of cloud microphysical processes on black carbon wet removal, global distributions, and radiative forcing"

_Atmospheric Chemistry and Physics, 2018_

## Referee Comment (RC1) · Anonymous Referee #1 · 25 Oct 2018

General comments

The authors developed a wet removal scheme that explicitly describes the influence of cloud processeson BC in CESM. Compared to the original scheme in CESM (i.e., MAM7), the improved wet scavenging scheme greatly reduces bias against HIPPO 1-4 aircraft observations. Finnaly, the authors calculated golbal total annual mean BC conversion rates among different phases, quantified the contributions of different cloud processes to the conversion rates, and evaluated the influences of these processes on BC distribution and direct raidtive forcing. Generally speaking, the paper is well written and documented, explanatory sections are interesting, and tables and graphics

are well constructed.As a result, I am recommending the paper be accepted with minor revisions. The few questions and comments I have are listed below in the specific comments to the authors.

Specific Comments

1. In the section 2.1, the parameterizations used in this study may be summaried in a table in order to make the paper more clear. In addition, the related information about the HIPPO campaigns (e.g., location, flight samples, time) is welcome.

2. One suggestion: Different cloud processes may affect the vertical profile of BC. Besides direct radiative forcing, I wonder that corresponding radiative heating rate profiles of BC caused by different cloud processes how to change??? I encourage the authors to perform related simulation in the current or further study.

Please also note the supplement to this comment:
https://www.atmos-chem-phys-discuss.net/acp-2018-900/acp-2018-900-RC1-supplement.pdf

---

## Referee Comment (RC2) · Anonymous Referee #2 · 26 Oct 2018

This study attempts to improve global model simulations of black carbon (BC) by resolving conversions of BC between cloud liquid, cloud ice, rain, snow, and interstitial air, which are tied to microphysics of clouds and precipitation. A series of sensitivity simulations show the relative importance of various cloud processes on BC distributions and associated radiative effect. The manuscript is a useful contribution to the scientific progress and the modeling methods employed are valid and clearly described. The manuscript is well written and is appropriate for publication in ACP.

Some minor revisions are suggested as follows:

(1) Page 1, Line 29. Change "Aerosol activation" to "Suppressing BC droplet activation

in clouds".

(2) Page 3, Line 27. Change "precipitated" to "settling".

(3) Page 4, Line 2. Change to "...and water vapor condenses".

(4) Page 4, Line 6. Cite Liu et al. (2011) and Wang et al. (2014) which are already listed in the References.

(5) Page 4, Line 21. Change "in an incomplete way" to "without considering all relevant microphysical processes".

(6) Pagr 5, Line 20. Change to "...and mass mixing ratios".

(7) Page 7, Line 10. Chnage to "BC aerosols are emitted".

(8) Page 7, Line 11-12. Add at the end of the sentence "although the aging time has been estimated in the range of .... (references).".

(9) Page 13, Figure 2. The arrow for evaporation of BC_snow is pointed toward "BC_phobic". Shouldn't is be extended (over BC_phobic) toward "BC_phlic"?

(10) Figure S1. Use "Cloud water conversion rate" or "Rate" to label the y-axis. The unit should be either "g/(m2 s)"(column integrated) or "(g/g)/s" (column averaged).

---

## Referee Comment (RC3) · Anonymous Referee #3 · 8 Nov 2018

The authors incorporate the impact of microphysical processes on the wet deposition of black carbon in the CAM5 global climate model. With this parameterization, they carry out a systematic evaluation of the importance of various microphysical processes on the distribution and radiative forcing of black carbon. Global distributions of black carbon remain highly uncertain, and this study provides a novel and substantial contribution towards the understanding a key piece of this complex problem. The paper is well-structured and the presentation is clear. I recommend minor revisions.

I have two main comments on how the conclusions can be better supported:

- Figure 2: authors have two full years of usable simulation data (2009-2010) and one partial year (2011). Do convection scavenging, aerosol activation, ice nucleation, evaporation, and below cloud scavenging dominate when the other years are considered?

- Figure 5: The colorbar saturates too quickly, making it hard to compare between simulations. In particular, it appears that convective scavenging (a) and cloud activation (b) are on par in the mid-latitudes, rather than the claim that the former dominates. Also, it is difficult to tell the changes in vertical profile, which is relevant for the section on radiative forcing.

I encourage the authors to frame their discussion on the direct radiative forcing (DRF) in the context of the major factors known to affect the direct radiative effect (e.g. in Equation 6.1 of Bond et al. (2013)): emissions, lifetime, absorption cross-section, and absorption efficiency. In particular, recent work suggests that there has been both an underestimate in emissions (e.g. Cohen and Wang (2014)) and overestimate in lifetime, and that the two factors act to cancel each other (Hodnebrog, Myhre, and Samset 2014). I agree that wet deposition is an important piece of constraining DRF; my concern is that a reader may walk away thinking that it is the only factor.

In the introduction, the authors may want to comment on the relative roles of transport vs removal, as in the introduction of Q. Wang et al. (2014).

**Specific comments**

- p1, line 26: I don't see significance tests, perhaps rephrase as 'largest impact'
- p1, line 29: do you mean "convection scavenging mainly increases the fraction of column BC below 5 km"?
- NO BERGERON and NO PRECIP EVAP are misspelt as NO BEGERON and NO PERCIP EVAP in some cases (eg p19 line 7, p21 line 21, Figure 7)
- p6, line 6: 'more accurately simulates' -> as compared to?

- p9, line 17: 'we turn off the impact of each cloud process on BC' -> I assume you mean that the changes in cloud processes do not affect the climate. Would be good to make clear.
- p14, line 6: do you mean 1.9 kg/s?
- please be consistent in use of abbreviations (e.g. fig in p15 vs Fig in p16, figure in p18 line 24 vs Figure in p18 line 1)

**References**

Bond, Tami C, Sarah J Doherty, DW Fahey, PM Forster, T Berntsen, BJ DeAngelo, MG Flanner, et al. 2013. "Bounding the Role of Black Carbon in the Climate System: A Scientific Assessment." *J. Geophys. Res.-Atmos.* 118 (11). Wiley Online Library: 5380–5552.

Cohen, Jason Blake, and Chien Wang. 2014. "Estimating Global Black Carbon Emissions Using a Top-down Kalman Filter Approach." *J. Geophys. Res.-Atmos.* 119 (1). Wiley Online Library: 307–23.

Hodnebrog, Øivind, Gunnar Myhre, and Bjørn H Samset. 2014. "How Shorter Black Carbon Lifetime Alters Its Climate Effect." *Nature Communications* 5. Nature Publishing Group: 5065.

Wang, Qiaoqiao, Daniel J Jacob, J Ryan Spackman, Anne E Perring, Joshua P Schwarz, Nobuhiro Moteki, Eloïse A Marais, Cui Ge, Jun Wang, and Steven RH Barrett. 2014. "Global Budget and Radiative Forcing of Black Carbon Aerosol: Constraints from Pole-to-Pole (Hippo) Observations Across the Pacific." *Journal of Geophysical Research: Atmospheres* 119 (1). Wiley Online Library: 195–206.

---

## Author Comment (AC2) · 3 Jan 2019

Response to Anonymous Referee # 2

(Note: Reviewer comments are listed in grey, and responses to reviewer comments are in black. Pasted text from the new version of the paper is in italics.)

The manuscript is a useful contribution to the scientific progress and the modeling methods employed are valid and clearly described. The manuscript is well written and is appropriate for publication in ACP. Some minor revisions are suggested as follows:

We greatly appreciate the reviewer for the detailed, valuable and constructive comments. The suggestions are extremely helpful to improving our manuscript.

Some minor revisions are suggested as follows:

1.  Page 1, Line 29. Change "Aerosol activation" to "Suppressing BC droplet activation in clouds".

    Thanks for this suggestion. We have changed the sentence according to the reviewer's comment:

    "*Suppressing BC droplet activation in clouds mainly decreases the fraction of column BC below 5 km whereas suppressing BC ice nucleation increase that above 10 km.*"

2.  Page 3, Line 27. Change "precipitated" to "settling".

    Thanks. We have changed the word:

    "*In mixed-phase clouds, observations have found that riming increase BC scavenging efficiency because settling ice crystals collect BC in the supercooled droplets of clouds at lower altitudes (Hegg et al., 2011).*"

3.  Page 4, Line 2. Change to "...and water vapor condenses".

    Thanks. We have changed the sentence:

    "*In addition to the riming, ice crystals can also grow through the Bergeron process—when water vapour pressure is supersaturated with respect to ice and undersaturated to liquid water, cloud droplets evaporate and water vapour condense onto ice crystals/snow.*"

4.  Page 4, Line 6. Cite Liu et al. (2011) and Wang et al. (2014) which are already listed in the References.

    Thanks for this comment. We have Cited Liu et al. (2011) and Wang et al. (2014):

*"Modelling studies suggest that the Bergeron process is important to the simulation of BC in the Arctic (Fan et al., 2012;Liu et al., 2011;Wang et al., 2014)."*

5. Page 4, Line 21. Change "in an incomplete way" to "without considering all relevant microphysical processes".

   Thanks. We have revised the sentence according to the reviewer's comment:

   *"Thus, most global models treat BC wet scavenging without considering all relevant microphysical processes (Textor et al., 2006;Wang et al., 2011;Croft et al., 2010;Qi et al., 2017)."*

6. Page 5, Line 20. Change to "...and mass mixing ratios".

   Thanks for pointing this out. We have revised the sentence:

   *"The stratiform cloud microphysics scheme used in CAM5 is double moment (Morrison and Gettelman, 2008), predicting number concentrations and mass mixing ratios of cloud particles as well as diagnosing number concentrations and mass of precipitation."*

7. Page 7, Line 10. Change to "BC aerosols are emitted".

   Thanks for pointing this out. We have changed the sentence:

   *"BC aerosols are emitted in combination of 80% hydrophobic $BC_{phobic}$ and 20% hydrophilic $BC_{philic}$."*

8. Page 7, Line 11-12. Add at the end of the sentence "although the aging time has been estimated in the range of .... (references)."

   Thanks for this comment. We have added references at the end of the sentence:

   *"Although the aging time has been estimated in the range of one hour to two weeks (Zhang et al., 2015;Fierce et al., 2015;Matsui, 2016) "*

9. Page 13, Figure 2. The arrow for evaporation of BC_snow is pointed toward "BC_phobic". Shouldn't is be extended (over BC_phobic) toward "BC_phlic"?

   Thanks for correcting this. We have modified Figure 2, as shown below:

[Figure]

**Figure 2. Global budget of BC conversion (kg/s) among interstitial hydrophobic BC (BC_phobic), interstitial hydrophilic BC (BC_philic), BC in cloud water (BC_water), BC in cloud ice (BC_ice), BC in rain (BC_rain), and BC in snow (BC_snow) due to different cloud processes and aging. The conversion rates shown in the figure represent global total values averaged for year 2009.**

10. Figure S1. Use "Cloud water conversion rate" or "Rate" to label the y-axis. The unit should be either "g/(m2 s)"(column integrated) or "(g/g)/s" (column averaged).

Thanks for pointing this out. We have modified the Figure S1 according to the reviewer's comment. Please see the Figure below:

[Figure]

**Figure S1. Cloud water column mean conversion rate due to Bergeron, riming, evaporation of cloud water and ice sedimentation, evaporation of cloud process, over four seasons (DJF, MAM, JJA, and SON).**

References:

Croft, B., Lohmann, U., Martin, R., Stier, P., Wurzler, S., Feichter, J., Hoose, C., Heikkilä, U., Donkelaar, A. v., and Ferrachat, S.: Influences of in-cloud aerosol scavenging parameterizations on aerosol concentrations and wet deposition in ECHAM5-HAM, Atmospheric Chemistry and Physics, 10, 1511-1543, 2010.

Fan, S. M., Schwarz, J. P., Liu, J., Fahey, D. W., Ginoux, P., Horowitz, L. W., Levy, H., Ming, Y., and Spackman, J. R.: Inferring ice formation processes from global-scale black carbon profiles observed in the remote atmosphere and model simulations, Journal of Geophysical Research: Atmospheres, 117, n/a-n/a, 10.1029/2012jd018126, 2012.

Fierce, L., Riemer, N., and Bond, T. C.: Explaining variance in black carbon's aging timescale, Atmospheric Chemistry and Physics, 15, 3173-3191, 10.5194/acp-15-3173-2015, 2015.

Hegg, D. A., Clarke, A. D., Doherty, S. J., and Ström, J.: Measurements of black carbon aerosol washout ratio on Svalbard, Tellus B, 63, 891-900, 2011.

Liu, J., Fan, S., Horowitz, L. W., and Levy, H.: Evaluation of factors controlling long-range transport of black carbon to the Arctic, Journal of Geophysical Research, 116, 10.1029/2010jd015145, 2011.

Matsui, H.: Black carbon simulations using a size-and mixing-state-resolved three-dimensional model: 2. Aging timescale and its impact over East Asia, Journal of Geophysical Research: Atmospheres, 121, 1808-1821, 2016.

Morrison, H., and Gettelman, A.: A New Two-Moment Bulk Stratiform Cloud Microphysics Scheme in the Community Atmosphere Model, Version 3 (CAM3). Part I: Description and Numerical Tests, Journal of Climate, 21, 3642-3659, 10.1175/2008jcli2105.1, 2008.

Qi, L., Li, Q., He, C., Wang, X., and Huang, J.: Effects of the Wegener–Bergeron–Findeisen process on global black carbon distribution, Atmospheric Chemistry and Physics, 17, 7459-7479, 2017.

Textor, C., Schulz, M., Guibert, S., Kinne, S., Balkanski, Y., Bauer, S., Berntsen, T., Berglen, T., Boucher, O., and Chin, M.: Analysis and quantification of the diversities of aerosol life cycles within AeroCom, Atmospheric Chemistry and Physics, 6, 1777-1813, 2006.

Wang, Q., Jacob, D. J., Fisher, J. A., Mao, J., Leibensperger, E., Carouge, C., Sager, P. L., Kondo, Y., Jimenez, J., and Cubison, M.: Sources of carbonaceous aerosols and deposited black carbon in the Arctic in winter-spring: implications for radiative forcing, Atmospheric Chemistry and Physics, 11, 12453-12473, 2011.

Wang, R., Tao, S., Shen, H., Huang, Y., Chen, H., Balkanski, Y., Boucher, O., Ciais, P., Shen, G., Li, W., Zhang, Y., Chen, Y., Lin, N., Su, S., Li, B., Liu, J., and Liu, W.: Trend in Global Black Carbon Emissions from 1960 to 2007, Environmental Science & Technology, 48, 6780-6787, 10.1021/es5021422, 2014.

Zhang, J., Liu, J., Tao, S., and Ban-Weiss, G. A.: Long-range transport of black carbon to the Pacific Ocean and its dependence on aging timescale, Atmospheric Chemistry and Physics, 15, 11521-11535, 10.5194/acp-15-11521-2015, 2015.

---

## Author Comment (AC1)

Response to Anonymous Referee # 1

(Note: Reviewer comments are listed in grey, and responses to reviewer comments are in black. Pasted text from the new version of the paper is in italics.)

The authors developed a wet removal scheme that explicitly describes the influence of cloud processes on BC in CESM. Compared to the original scheme in CESM (i.e., MAM7), the improved wet scavenging scheme greatly reduces bias against HIPPO 1-4 aircraft observations. Finally, the authors calculated global total annual mean BC conversion rates among different phases, quantified the contributions of different cloud processes to the conversion rates, and evaluated the influences of these processes on BC distribution and direct radiative forcing. Generally speaking, the paper is well written and documented, explanatory sections are interesting, and tables and graphics are well constructed. As a result, I am recommending the paper be accepted with minor revisions. The few questions and comments I have are listed below in the specific comments to the authors.

We really appreciate the thoughtful and valuable comments from the reviewer. These comments substantially help to improve our manuscript by addressing these issues.

Specific Comments

1.  In the section 2.1, the parameterizations used in this study may be summaried in a table in order to make the paper more clear. In addition, the related information about the HIPPO campaigns (e.g., location, flight samples, time) is welcome.

    Thanks for this great suggestion! In order to make the paper easy to read, we have inserted Table 1 to summarize all equations. Meanwhile, we have provided more background information of HIPPO campaigns in section 2.4:

"*In order to evaluate our new parameterization, we compare model simulation results with aircraft measurements from HIAPER Pole-to-Pole Observation (HIPPO). The HIPPO observations provide extensive vertical profiles of 26 species from the surface to 14 km above the remote Pacific, spanning from 85°N to 67°S. Five deployments were conducted in periods of 8–30 January 2009, 31 October – 22 November 2009, 24 March – 16 April 2010, 14 June – 11July 2011, 9 August – 9 September 2011(Wofsy, 2011). BC Particles were measured using a single-particle soot photometer (SP2)(Schwarz et al., 2010). Because the aircraft both ascends and descends along each flight track, HIPPO generates vertical profiles of BC concentrations.*"

**Table 1. Cloud processes associated with our improved BC wet removal parameterization, BC conversion along with each cloud process, and corresponding conversion rate as described by Equations (1)-(11).**

| PROCESS | BC CONVERSION | BC CONVERSTION RATE |
|---|---|---|
| Cloud activation | $BC_{philic}$ to $BC_{water}$ | $k_{philic \to water} = \dfrac{CDNC}{N_{aerosol-CCN}}$ |
| Ice nucleation | $BC_{philic}$ to BC ice | $k_{philic \to ice} = \dfrac{ICNC}{N_{aerosol-IN}}$ |
| Contact freezing, immersion freezing, homogeneous freezing, riming splintering | $BC_{water}$ to $BC_{ice}$ | $k_{water \to ice} = \dfrac{CONTACT + IMMERSION + HOMO + SPLINT}{Q_{liq}}$ |
| Melting | $BC_{ice}$ to $BC_{water}$ | $k_{ice \to water} = \dfrac{MELT}{Q_{ice}}$ |
| Evaporation of the cloud, the Bergeron process and evaporation of cloud water sedimentation | $BC_{water}$ to $BC_{philic}$ | $k_{water \to philic} = \dfrac{EVP\_CLOUD + BERG + EVP\_CSEDI}{Q_{liq}}$ |
| sublimation of cloud ice sedimentation | $BC_{ice}$ to $BC_{philic}$ | $k_{ice \to philic} = \dfrac{EVP\_ISEDI}{Q_{ice}}$ |
| Autoconversion and accretion | $BC_{water}$ to $BC_{rain}$s | $k_{water \to rain} = \dfrac{PRAO + PRCO}{Q_{liq}}$ |
| Collision and coalescence | $BC_{ice}$ to $BC_{snow}$ | $k_{ice \to snow} = \dfrac{PRAIO + PRCIO}{Q_{liq}}$ |
| Riming | $BC_{water}$ to $BC_{snow}$ | $k_{water \to snow} = \dfrac{RIMING}{Q_{liq}}$ |
| Deep and shallow convection scavenging | Deposition of $BC_{phobic}$ | $k_{phobic \to convection} = \dfrac{RRDP + RRSH}{Q_{liq} + Q_{ice}}$ |
| Deep and shallow convection scavenging | Deposition of $BC_{philic}$ | $k_{philic \to convection} = \dfrac{RRDP + RRSH}{Q_{liq} + Q_{ice}}$ |

**2.** One suggestion: Different cloud processes may affect the vertical profile of BC. Besides direct radiative forcing, I wonder that corresponding radiative heating rate profiles of BC caused by different cloud processes how to change??? I encourage the authors to perform related simulation in the current or further study.

Thanks for the reviewer's valuable suggestion. We will address this and perform related simulations in our future work to see how the influence of cloud processes on aerosols would modify the radiative heating rate.

References:

Schwarz, J., Spackman, J., Gao, R., Watts, L., Stier, P., Schulz, M., Davis, S., Wofsy, S. C., and Fahey, D.: Global‑scale black carbon profiles observed in the remote atmosphere and compared to models, Geophysical Research Letters, 37, 2010.

Wofsy, S. C.: HIAPER Pole-to-Pole Observations (HIPPO): fine-grained, global-scale measurements of climatically important atmospheric gases and aerosols, Philosophical Transactions of the Royal Society of London A: Mathematical, Physical and Engineering Sciences, 369, 2073-2086, 2011.

---

## Author Comment (AC3)

Response to Anonymous Referee # 3

(Note: Reviewer comments are listed in grey, and responses to reviewer comments are in black. Pasted text from the new version of the paper is in italics.)

The authors incorporate the impact of microphysical processes on the wet deposition of black carbon in the CAM5 global climate model. With this parameterization, they carry out a systematic evaluation of the importance of various microphysical processes on the distribution and radiative forcing of black carbon. Global distributions of black carbon remain highly uncertain, and this study provides a novel and substantial contribution towards the understanding a key piece of this complex problem. The paper is well-structured and the presentation is clear. I recommend minor revisions.

We greatly appreciate the reviewer's thorough and constructive review. We believe revising the paper according to the reviewer's comments has considerably improved the paper. We have merged all of the suggestions into the revised manuscript. Please see our response to each comment below:

I have two main comments on how the conclusions can be better supported:

1.  Figure 2: authors have two full years of usable simulation data (2009-2010) and one partial year (2011). Do convection scavenging, aerosol activation, ice nucleation, evaporation, and below cloud scavenging dominate when the other years are considered?

    Thanks for bringing up this important issue. We have extended our simulation to the end of 2011. Following the reviewer's comment, we plot the zonal mean fractional changes averaged in year 2010 and 2011. The figure below is very similar to figure 2, which shows the changes for year 2009. Simulations in year 2010 and 2011 also indicate that convection scavenging, aerosol activation, ice nucleation, evaporation, and below cloud scavenging dominate BC vertical distribution.

[Figure]

**Figure. Zonal mean fractional changes (unitless) in BC concentrations averaged in year 2010 and year 2011 relative to BASE in eight sensitivity simulations when the influence of one cloud process on BC is**

**turned off. The sensitivity simulations are described in section 2.3, including (a) NO CONVECTION (no convection scavenging), (b) NO CCN (no cloud activation), (c) NO IN (no ice nucleation), (d) NO RIMING (no riming), (e) NO BELOW CLOUD (no below cloud scavenging), (f) NO BERGERON (no Bergeron process), (g) NO CLOUD EVAP (no evaporation of cloud water/ice sedimentation), and (h) NO PRECIP EVAP (no evaporation of rain/snow).**

2. Figure 5: The color bar saturates too quickly, making it hard to compare between simulations. In particular, it appears that convective scavenging (a) and cloud activation (b) are on par in the mid-latitudes, rather than the claim that the former dominates. Also, it is difficult to tell the changes in vertical profile, which is relevant for the section on radiative forcing.

Thanks for this great suggestion. Following the reviewer's comment, we have modified the color table in Figure 5 to more clearly demonstrate the results. As the reviewer mentioned, the effects of convective scavenging and activation are on par in the mid-latitudes, however, the influence of NO CONVECTION on global burden is the largest among all sensitivity simulations. Therefore, we conclude that convection dominates BC burden. We also have plotted Figure 6 to show the influences of different cloud processes on BC vertical distribution.

[Figure]

**Figure 5. Annual zonal mean fractional changes (unitless) for year 2009 in BC concentrations relative to BASE in eight sensitivity simulations when the influence of one cloud process on BC is turned off. The sensitivity simulations are described in section 2.3, including (a) NO CONVECTION (no convection scavenging), (b) NO CCN (no cloud activation), (c) NO IN (no ice nucleation), (d) NO RIMING (no riming), (e) NO BELOW CLOUD (no below cloud scavenging), (f) NO BERGERON (no Bergeron process), (g) NO CLOUD EVAP (no evaporation of cloud water/ice sedimentation), and (h) NO PRECIP EVAP (no evaporation of rain/snow).**

[Figure]

**Figure 6. Fraction of global BC burden at four altitude bands for year 2009 in the BASE simulation using our new wet removal scheme described in section 2.2, and sensitivity simulations when the influence of each cloud process on BC is turned off. The sensitivity simulations are described in section 2.3, including NO CONVECTION (no convection scavenging), NO CCN (no cloud activation), NO IN (no ice nucleation), NO RIMING (no riming), NO BELOW CLOUD (no below cloud scavenging), NO BERGERON (no Bergeron process), NO CLOUD EVAP (no evaporation of cloud water/ice sedimentation), NO PRECIP EVAP (no evaporation of rain/snow).**

I encourage the authors to frame their discussion on the direct radiative forcing (DRF) in the context of the major factors known to affect the direct radiative effect (e.g. in Equation 6.1 of Bond et al. (2013)): emissions, lifetime, absorption cross-section, and absorption efficiency. In particular, recent work suggests that there has been both an underestimate in emissions (e.g. Cohen and Wang (2014)) and overestimate in lifetime, and that the two factors act to cancel each other (Hodnebrog, Myhre, and Samset 2014). I agree that wet deposition is an important piece of constraining DRF; my concern is that a reader may walk away thinking that it is the only factor. In the introduction, the authors may want to comment on the relative roles of transport vs removal, as in the introduction of Q. Wang et al. (2014).

Excellent point. I agree that our study only consider wet removal as a factor contribute to uncertainty of DRF, which may mislead our reader. We reframe the beginning of section 5 as below :

"*Emissions, lifetime, absorption cross-section, and absorption efficiency can all affect BC DRF (Bond et al., 2013). The total impacts of uncertainties in these processes on BC DRF estimates is complex. For instance, recent studies suggest that there has been both an underestimate in emissions (e.g.Cohen and Wang (2014)) and an overestimate in lifetime, and that the two factors act to cancel each other (Hodnebrog et al., 2014). In this study, we only focus on how cloud processes influence BC DRF via altering BC wet removal.*"

Meanwhile, we also added a paragraph in the introduction to compare the relative importance of emission, transport, dry deposition, and wet removal in controlling BC concentrations:

*"The inter-model discrepancies and disagreement between models and measurements reflect uncertainties in emissions, transport, dry deposition, and wet scavenging of BC simulation. The uncertainties in BC concentrations over source regions are mainly contributed by errors in emission inventories. Fu et al. (2012) and Leibensperger et al. (2012) suggest that emission inventories lead to normalized mean bias of less than 2 against observations over source regions. Using inert 222Rn as a tracer, previous studies show that pollution transport in three dimensional models is fairly well constrained with observations; seasonality and magnitude of profiles 222Rn vertical profiles are captured by the models (Jacob et al., 1997;Stockwell and Chipperfield, 1999). Dry and wet deposition are the sinks of BC. Previous literature suggests that global total wet deposition is 3-6 times larger than dry deposition (Jurado et al., 2008;Huang et al., 2010;Zhang et al., 2015). In the remote troposphere, wet scavenging is considered as the most important source of BC simulation uncertainties (Koch et al., 2009;Schwarz et al., 2010;Croft et al., 2010;Liu et al., 2011;Wang et al., 2014). "*

Specific comments

1. p1, line 26: I don't see significance tests, perhaps rephrase as 'largest impact'

   Thanks for this comment. We have changed the phrase:

   "*Convective scavenging is found to have the largest impact on BC concentrations at mid-altitudes over the tropics and even globally.*"

2. p1, line 29: do you mean "convection scavenging mainly increases the fraction of column BC below 5 km"?

   Thanks. Based on Figure 6, convection scavenging greatly influences the fraction of BC at all altitude bands. We have change the sentence to make this point clearer:

   "*As for BC vertical distributions, convective scavenging greatly influences BC fractions at different altitudes.*"

3. NO BERGERON and NO PRECIP EVAP are misspelt as NO BEGERON and NO PERCIP EVAP in some cases (eg p19 line 7, p21 line 21, Figure 7)

   Thanks for catching this mistake. We have modified the sentences and Figure 7, as shown below:

   "*Therefore, NO BERGERON decreases BC concentrations in the Arctic relative to BASE (Fig. 5(f)).*"

[Figure]

**Figure 7. Change in global radiative forcing of BC estimated by sensitivity simulations relative to BASE for year 2009. The sensitivity simulations are described in section 2.3, including (a) NO CONVECTION (no convection scavenging), (b) NO CCN (no cloud activation), (c) NO IN (no ice nucleation), (d) NO RIMING (no riming), (e) NO BELOW CLOUD (no below cloud scavenging), (f) NO BERGERON (no Bergeron process), (g) NO CLOUD EVAP (no evaporation of cloud water/ice sedimentation), and (h) NO PRECIP EVAP (no evaporation of rain/snow).**

4. p6, line 6: 'more accurately simulates' -> as compared to? 1

Thanks for pointing this out. The shallow convection that we used was compared to Hack et al. (1994) shallow convection scheme in CAM3 and CAM4. We revised the sentence:

"*Shallow convection is treated with a parameterization developed by Park and Bretherton (2009) that computes vertical velocity and fractional area of convection, and more accurately simulates spatial distribution of shallow column activity, as compared to Hack (1994) shallow convection scheme in CAM3 and CAM4.*"

5. p9, line 17: 'we turn off the impact of each cloud process on BC' -> I assume you mean that the changes in cloud processes do not affect the climate. Would be good to make clear.

Good suggestion. We have added the following sentence in section 2.3 to make it clear.

[revised manuscript text omitted]